# Evaluating the Impact of Medical Image Reconstruction on Downstream AI Fairness and Performance

**Matteo Wohlrapp**[1,2]               MATTEO.WOHLRAPP@CDTM.DE
**Niklas Bubeck**[2,3]                NIKLAS.BUBECK@TUM.DE
**Daniel Rueckert**[2,3,4]              DANIEL.RUECKERT@TUM.DE
**William Lotter**[1,5]          LOTTERB@DS.DFCI.HARVARD.EDU

[1]*Department of Data Science, Dana-Farber Cancer Institute, Boston, MA, USA*

[2]*AI in Medicine, Technical University of Munich, Munich, Germany*

[3]*Munich Center for Machine Learning (MCML), Munich, Germany*

[4]*Department of Computing, Imperial College London, London, UK*

[5]*Harvard Medical School, Boston, MA, USA*

**Editors:** Accepted for publication at MIDL 2026

## Abstract

AI-based image reconstruction models are increasingly deployed in clinical workflows to improve image quality from noisy data, such as low-dose X-rays or accelerated MRI scans. However, these models are typically evaluated using pixel-level metrics like PSNR, leaving their impact on downstream diagnostic performance and fairness unclear. We introduce a scalable evaluation framework that applies reconstruction and diagnostic AI models in tandem, which we apply to two tasks (classification, segmentation), three reconstruction approaches (U-Net, GAN, diffusion), and two data types (X-ray, MRI) to assess the potential downstream implications of reconstruction. We find that conventional reconstruction metrics poorly track task performance, where diagnostic accuracy remains largely stable even as reconstruction PSNR declines with increasing image noise. Fairness metrics exhibit greater variability, with reconstruction sometimes amplifying demographic biases, particularly regarding patient sex. However, the overall magnitude of this additional bias is modest compared to the inherent biases already present in diagnostic models. To explore potential bias mitigation, we adapt two strategies from classification literature to the reconstruction setting, but observe limited efficacy. Overall, our findings emphasize the importance of holistic performance and fairness assessments throughout the entire medical imaging workflow, especially as generative reconstruction models are increasingly deployed.

**Keywords:** Fairness, Image Reconstruction, GANs, Diffusion Models

## 1. Introduction

AI-based image reconstruction is an increasingly integral component of clinical workflows. These approaches are designed to enhance the quality of noisy medical images such as low-dose X-rays or faster-sampled MRIs, ultimately generating new medical images by imputing patterns learned from the training datasets (Ahishakiye et al., 2021). Notably, there are now over 80 FDA-cleared devices based on this approach (Singh et al., 2025), whose generated images are ultimately interpreted by clinicians.

Traditionally, reconstruction model performance has been evaluated using pixel-level image metrics such as PSNR. However, these metrics provide an incomplete picture, as they

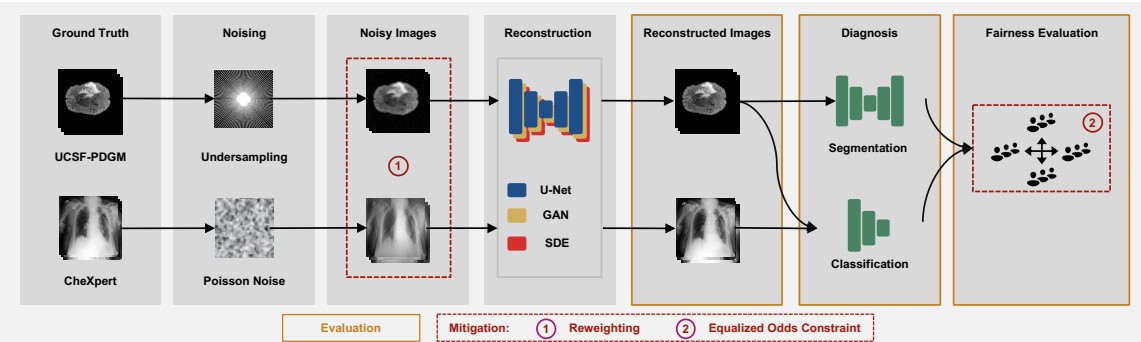

Figure 1: Combined pipeline for downstream bias evaluation and mitigation in medical image reconstruction. MRI and X-ray images undergo realistic simulated degradation and are subsequently reconstructed with three approaches before serving as input to downstream prediction models. Reconstruction quality, downstream performance, and fairness are evaluated. Subsequently, two bias mitigation strategies are applied exclusively during reconstruction fine-tuning.

do not reflect the impact of reconstructed images on subsequent clinical tasks. This gap raises a key unresolved question: *How does AI-based reconstruction influence downstream clinical performance and, in particular, fairness?* The latter is especially important to assess given the risk of generative models in encoding biases (Saumure et al., 2025; Ruggeri and Nozza, 2023; Luccioni et al., 2023; Mehrabi et al., 2021). While some smaller-scale studies have involved clinician review of AI-reconstructed images (Feuerriegel et al., 2023; Lee et al., 2024), this approach is not scalable, especially when investigating nuanced performance differences across subgroups.

In this work, we assess the downstream implications of AI-based reconstruction through an evaluation framework that leverages reconstruction and classification/segmentation AI models applied in tandem. The framework provides a scalable approach to understand how reconstruction errors propagate, while also simulating a realistic clinical scenario as both reconstruction and diagnostic models are increasingly deployed in medical workflows. We apply this framework across three reconstruction approaches (U-Net, GAN, diffusion), two imaging domains (MRI, X-ray), and two tasks (classification and segmentation). We additionally propose and evaluate bias mitigation techniques tailored to reconstruction models. Our findings highlight differences in trends between image metrics and diagnostic accuracy, and the potential of reconstruction models to shift demographic biases.

## 2. Related Work

**Reconstruction Models in Medical Imaging:** Medical image reconstruction is a popular AI application due to its promise in increasing image quality while facilitating lower radiation doses and faster scanning times (Ahishakiye et al., 2021; Diab and Lotter, 2025). Given pairs of noisy (i.e., undersampled/lower dose) and original images, these models are trained to reconstruct the original from the noisy image. Recently, unsupervised methods

have also been developed that do not require paired original and noisy images (Chen et al., 2023; Sultan et al., 2025; Chen et al., 2026). Variations of the U-Net (Ronneberger et al., 2015) are commonly used as the neural network architecture for reconstruction models. In addition to standard losses like mean-squared error (MSE), GAN and diffusion-based approaches are common in the field (Bousse et al., 2024; Heckel et al., 2024).

**Fairness Analysis in Medical Imaging:** Research on bias in AI-driven healthcare spans various medical domains, with medical imaging receiving considerable attention. In classification tasks, biases are typically revealed by comparing performance across subgroups. Studies cover various imaging modalities, including brain MRI (Stanley et al., 2022; Ioannou et al., 2022), chest X-rays (Seyyed-Kalantari et al., 2021; Glocker et al., 2023; Yang et al., 2024; Lotter, 2024), dermatology images (Chiu et al., 2024; Groh et al., 2021), and retinal images (Burlina et al., 2021). They address sensitive attributes such as sex (Stanley et al., 2022), age (Seyyed-Kalantari et al., 2021), race (Seyyed-Kalantari et al., 2021), and skin tone (Kinyanjui et al., 2020), evaluating disparities using performance metrics such as Area Under the Curve (AUC) (Seyyed-Kalantari et al., 2021), or more dedicated fairness criteria (Yuan et al., 2023). In segmentation, studies have assessed segmentation performance under varying demographic distributions, such as by race and sex representation in training datasets (Ioannou et al., 2022; Lee et al., 2022; Puyol-Antón et al., 2022).

**Fairness Analysis of Reconstruction Models:** Reconstruction model performance is typically measured using image quality metrics such as Peak Signal-to-Noise Ratio (PSNR) and Structural Similarity Index Measure (SSIM). Recent studies assessing subgroup biases primarily rely on these metrics, examining how image quality varies across demographic subgroups. For instance, Du et al. (2023b) investigated fairness in deep learning-based brain MRI reconstruction, highlighting disparities in image reconstruction quality across different demographic groups using PSNR and SSIM. Similarly, Sheng et al. (2024) explored fairness challenges and potential solutions in ultrasound computed tomography, identifying significant disparities in reconstruction performance linked to subgroup attributes. With limited available literature, bias evaluation in reconstruction models is an emerging area of research for which there is a need to study the implications of image reconstruction on downstream tasks.

**Bias Mitigation:** In classification, substantial efforts have focused on developing bias mitigation strategies. Data-centric approaches directly modify training datasets, employing methods such as data redistribution (Oguguo et al., 2023), differentiable resampling techniques (Li and Vasconcelos, 2019), harmonization of datasets (Bissoto et al., 2019), and synthetic generation of diverse samples (Wang et al., 2024). Additionally, methods like Just Train Twice (JTT) target misclassified instances to implicitly mitigate subgroup biases without explicit annotations (Liu et al., 2021).

Representation-level strategies aim to learn unbiased feature representations through explicit disentanglement. Techniques include variational autoencoders (Creager et al., 2019), orthogonal disentanglement methods enforcing independence between sensitive attributes and task-specific features (Sarhan et al., 2020; Deng et al., 2023; Chiu et al., 2024; Du et al., 2023a), and group-adaptive architectures employing demographic-specific attention mechanisms (Gong et al., 2020).

Optimization-level methods integrate fairness constraints into model training via adversarial learning, fairness-specific loss functions, or specialized training regimens. Adversarial methods discourage encoding protected attributes (Zhang et al., 2018; Adeli et al., 2019; Kim et al., 2019; Wang et al., 2020), distributionally robust optimization (Group DRO) targets worst-case subgroup performance (Sagawa et al., 2020), and fairness-specific constraints can be incorporated directly into training (Marcinkevics et al., 2022). Post-processing methods adjust model outputs after training, employing techniques such as calibration and pruning (Wu et al., 2022).

While prior studies have focused mainly on bias mitigation in classification tasks, there remains a critical need to assess analogous strategies for image reconstruction.

## 3. Methods

Our framework, visualized in Figure 1, encompasses image denoising, downstream task evaluation, fairness assessment, and bias mitigation for medical image reconstruction. The framework uses classification and segmentation models to estimate the effect of reconstruction on downstream task performance and fairness. Additionally, mitigation strategies are applied exclusively at the reconstruction stage to determine their ability to reduce downstream biases without retraining diagnostic models.

### 3.1. Datasets

We apply our framework to public datasets from two distinct imaging domains:

**MRI:** UCSF-PDGM includes 501 pre-operative glioma MRI exams from patients with diffuse glioma, along with tumor masks and labels for subtype and grade (Calabrese et al., 2022). We use the T2-weighted FLAIR volumes for all analyses.

**X-Ray:** CheXpert comprises 224,316 radiographs from 65,240 patients annotated for 14 thoracic findings (Irvin et al., 2019), of which we use 12 (excluding "Support Devices" and "No Findings" to focus on disease pathologies).

We use a 70/10/20 train/validation/test split stratified by patient for both datasets. For CheXpert, the training set is further divided into non-overlapping sets for reconstruction and classification model training, with percentages of 70/30, respectively. For UCSF-PDGM, the same training data is for both tasks given smaller sample size. Group-wise fairness is assessed for age (dichotomized at the dataset median), sex, and self-reported race (unavailable for UCSF-PDGM). Detailed attribute distributions are reported in Tables 5 and 6 in the Appendix.

### 3.2. Noising Process

We simulate realistic acquisition degradations as follows:

**MRI:** $k$-space data is masked with radial undersampling patterns (Feng, 2022) at acceleration factors 4, 8, and 16, where higher acceleration means greater undersampling (see Appendix A).

**X-Ray:** Standard-dose images are Radon-projected to sinogram space, bow-tie filtered, and corrupted with Poisson noise parameterized by photon count (100,000, 10,000, 3,000), with lower photon count yielding more noise (Gibson et al., 2023).

These ranges approximate realistic acquisition conditions, with examples in the Appendix (Figures 6, 7, 8, and 9).

### 3.3. Models

We employ three reconstruction models alongside task-specific diagnostic models. Additional information on the compute infrastructure and model hyperparameters can be found in the Appendix.

**Reconstruction:** To cover deterministic, adversarial, and diffusion regimes, we train from scratch a standard U-Net (Ronneberger et al., 2015) with MSE loss, a Pix2Pix GAN (Isola et al., 2017), and a Stochastic Differential Equations (SDE)-based diffusion model (Luo et al., 2023) for each dataset. We note that the GAN and diffusion models also use a U-Net as the model architecture, but are based on a different training paradigm.

**Diagnostic:** For classification on UCSF-PDGM, an ImageNet-initialized ResNet50 (He et al., 2015) was trained separately to predict WHO grade and tumor type. The model is trained at the slice-level, and at testing, volume-level predictions are performed individually on each slice and then aggregated using the median. For CheXpert classification, a single ImageNet-initialized DenseNet model (Huang et al., 2017) was trained to jointly predict the 12 findings following Cohen et al. (2021). For segmentation on UCSF-PDGM, we use an ImageNet-initialized U-Net. Segmentation is not evaluated on CheXpert due to the absence of masks. All downstream models are trained on the original, non-degraded images.

### 3.4. Performance and Fairness Evaluation

Reconstruction quality is measured by PSNR. Downstream performance uses AUROC for classification and Dice for segmentation. For classification fairness, we report the worst-case Equalized Odds (EODD) (Hardt et al., 2016) difference between groups:

$$max_{i,j}|P(\hat{Y} = 1 \mid Y = y, A = a_i)$$
$$- P(\hat{Y} = 1 \mid Y = y, A = a_j)|, \quad \forall y \in \{0, 1\},$$
$$\forall \quad \text{attribute A} \in \mathcal{A}, \quad \text{subgroups} \quad a_i.$$

To compute this metric, model predictions are binarized using a balanced threshold selected to achieve approximately equal sensitivity and specificity in the validation split. Equality of Opportunity (EOP) results are also reported in the Appendix (Figures 14 and 15).

For segmentation fairness, we adapt the Skewed-Error Ratio (SER) (Siddiqui et al., 2024) to Dice:

$$SER_A = \frac{\max_i(1 - \text{Dice}_{a_i})}{\min_j(1 - \text{Dice}_{a_j})}, \quad a_i \in A, \quad A \in \mathcal{A}$$

Results using an unnormalized Dice difference are also provided in the Appendix (Figure 15).

Statistical comparisons of subgroup fairness differences were performed using boot-strapped estimates with 1,000 iterations. Bootstrap-derived p-values were used to determine statistical significance with a two-sided $p < 0.05$.

### 3.5. Bias Mitigation

We adapt two bias mitigation strategies that were originally proposed for classification models. Each approach involves fine-tuning only the reconstruction models after the original training described above. The differentiable equalized-odds approach also relies on using the reconstruction and classification models applied in tandem, but the classification network is frozen to exclusively assess the potential for bias mitigation at the reconstruction stage.

**Sample Reweighting:** A weighted sampler draws each example with inverse joint sub-group frequency during fine-tuning, ensuring that each subgroup (and combination thereof across attributes) is represented with the same frequency. The reconstruction model is fine-tuned using the corresponding original reconstruction loss.

**Differentiable Equalized-Odds:** For reconstruction output $\hat{x} = f(x)$ and classifier output $\hat{y} = g(\hat{x})$ we minimize: $\mathcal{L}_{\text{EODD}} = \ell_{\text{rec}}(\hat{x}) + \lambda_{\text{fair}} \text{EMA}\big(\ell_{\text{BCE}}(\hat{y}) + \text{EODD}^2\big)$, where $\ell_{\text{rec}}$ represents the original reconstruction loss for the model, $\ell_{\text{BCE}}$ represents the binary cross-entropy loss for the frozen classifier, EMA represents an exponential moving average, and EODD represents a differentiable Equalized Odds constraint inspired by Marcinkevics et al. (2022). Specifically, we use the maximum EODD difference of any subgroup as defined above and compute it via soft predictions: $\tilde{y} = \sigma\big((\hat{y}) - \tau)/T\big)$, where the threshold $\tau$ and temperature $T$ are set at 0.5 and 0.3, respectively. One loss is computed across all sensitive attributes (i.e., the max EODD over age, sex, and race). In the Appendix, we show that minimizing $\text{EODD}^2$ between subgroups corresponds to minimizing their covariance.

**Code:** Available at https://github.com/lotterlab/reconstruction_evaluation

## 4. Results

We first evaluate the impact of reconstruction on downstream task performance before analyzing fairness and the effectiveness of mitigation techniques.

### 4.1. Impact of Reconstruction on Task Performance

Figure 2 summarizes downstream performance as a function of reconstruction noise. We report segmentation Dice for UCSF-PDGM and the mean AUROC across the 12 CheXpert pathologies. For clarity, the y-axes for PSNR and the task metrics are normalized to the same percentage range. Across all experiments, diagnostic performance remains largely un-changed, even though PSNR decreases substantially with increasing noise. Specifically, the Dice score for UCSF-PDGM segmentation varies by no more than $\sim 3\%$ across noise condi-tions, and the mean CheXpert AUROC fluctuates by only $1\%$. In contrast, PSNR decreases by over 10 dB ($26\%$) for UCSF-PDGM and by $\sim 3$ dB ($9\%$) for CheXpert. Analogous re-sults for UCSF-PDGM classification are presented in the Appendix (Figure 10), where the same pattern—substantial PSNR loss but minimal impact on task performance—holds for

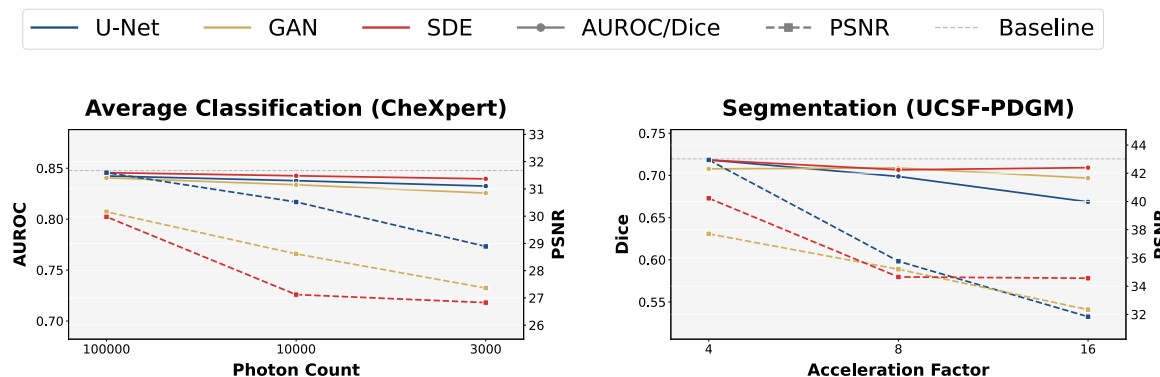

Figure 2: Downstream performance and PSNR at varying noise levels. Axes for PSNR and task performance are scaled to comparable percentage ranges. Although PSNR declines as noise increases, task performance remains stable. Baseline indicates performance on original images.

all three reconstruction models. Using SSIM instead of PSNR as the reconstruction metric also shows similar trends (Appendix Figure 18).

A closer look at CheXpert reveals a mild dependence on baseline task difficulty: pathologies with lower initial AUROC show slightly larger declines. For example, consolidation remains stable with U-Net reconstruction AUROC at 0.91, whereas lung lesion drops from 0.79 to 0.77 as noise increases (see Appendix Table 7 for more details).

### 4.2. Impact of Reconstruction on Fairness

Although aggregate task performance is largely unaffected, reconstruction could still alter relative performance across demographic subgroups. To test this possibility, we evaluated fairness on the downstream models using acceleration factor 8 for UCSF-PDGM and a photon count of 10,000 for CheXpert, representing the middle noise levels.

|  | Sex | Age | Race |
|---|---|---|---|
| **Classification** | $0.05 \pm 0.07$ | $0.17 \pm 0.08$ | $0.07 \pm 0.03$ |
| **Segmentation** | 1.10 | 1.22 | – |

Table 1: Baseline fairness of the classifiers (EODD) and the segmentation model (SER) for different sensitive attributes. For classification, mean and s.d. are reported across all classification tasks. Segmentation corresponds to UCSF-PDGM performance. Sex exhibits the lowest baseline bias.

Fig. 3 displays the distribution of bias shifts when reconstructed images replace the original inputs. To provide a global overview, the histogram represents the bias shifts across all tasks, pathologies, and reconstruction models. As the diagnostic models exhibit bias on the original inputs (Table 1), the bias shifts with reconstruction are plotted on a percentage scale compared to the original bias to highlight the relative effects. We find

**Distribution of Bias Changes**

Figure 3: Distribution of bias changes (percent change compared to original images) across all reconstruction models, datasets, and tasks, stratified by sensitive attribute. The vertical lines mark the medians. Most shifts cluster near zero, but sex shows a broader positive tail. Separate plots by sensitive attribute are contained in Fig. 20.

that the mode of these shifts is centered around zero, indicating little bias change in most instances. However, there is a noticeable tail towards positive bias changes, especially for sex, which exhibits a median increase of 24%. This is partly attributable to sex having a lower baseline bias than age and race (Table 1).

The bias changes for each pathology and model are provided in Figures 4 and 5 (represented by the "Reconstruction" value in each plot). For UCSF-PDGM, no significant fairness deviations were observed when using the reconstructed images compared to the original images for either the segmentation or classification tasks (Figure 5). CheXpert shows more frequent significant shifts. Out of the 36 combinations (12 pathologies x 3 reconstruction models), there were 8 significant changes for sex (all in the positive direction) and 12 significant changes for age (4 in the positive direction). Due to large error bars, there were 0 significant changes for race, but alternative analysis which excluded subgroups with small sample sizes did reveal some significant changes (Appendix A). Overall, the pathology-level findings support the histogram trend with a slight bias increase for sex and a slight decrease for age. The absolute magnitude of the effects were generally modest; however, some are of the order of a 0.05 change in EODD, corresponding to a 5% difference in sensitivity/specificity, which is meaningful at the population level. Across reconstruction methods, the GAN and SDE-based models exhibited smaller bias shifts than the traditional U-Net (Table 2).

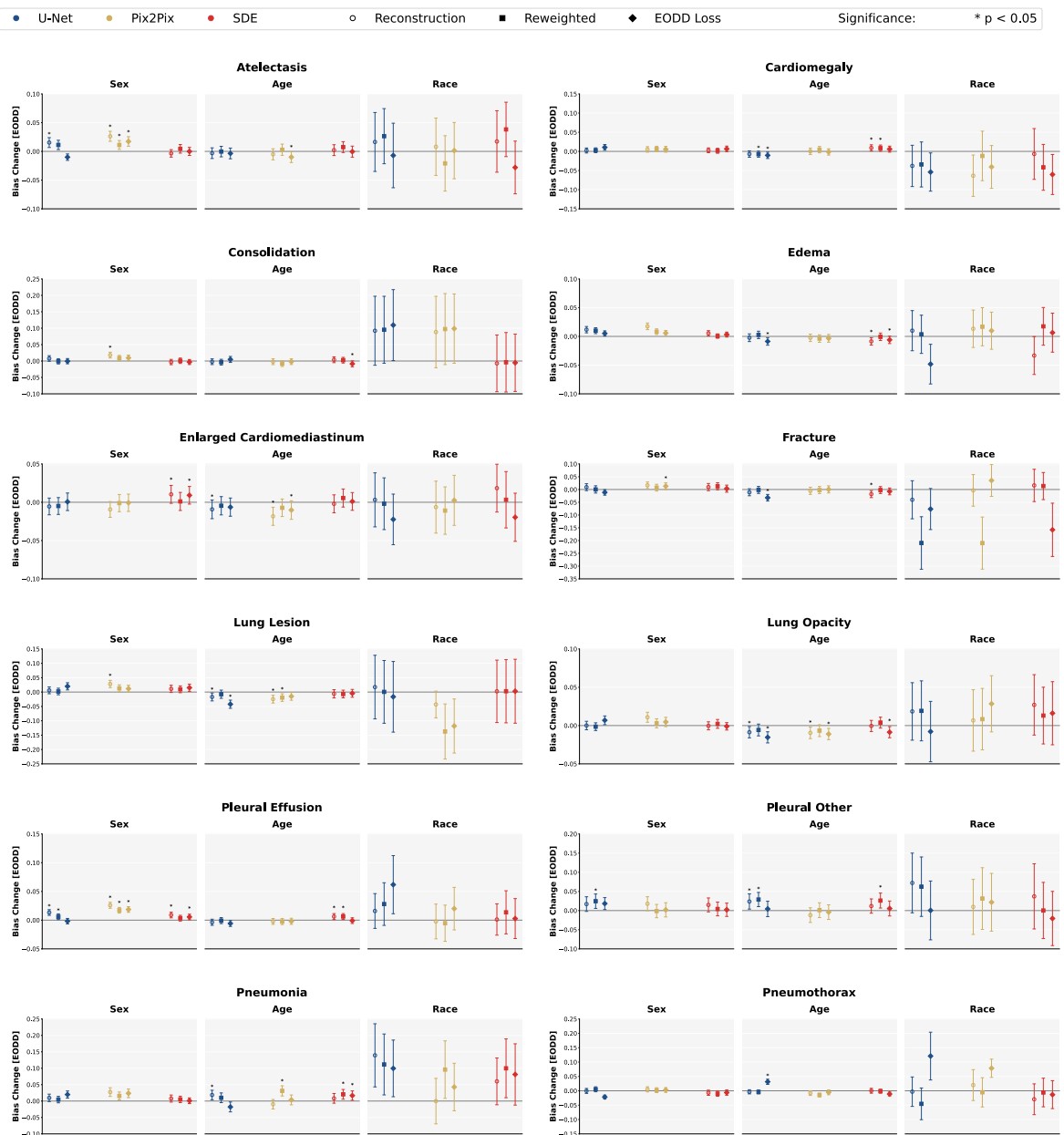

Figure 4: Equalized odds bias change pre- and post-mitigation compared to predictions on original images for CheXpert. Pre-mitigation ("Reconstruction"), bias tends to increase slightly for sex; race exhibits high variance. Bias tends to decline slightly post-mitigation. Error bars represent standard deviation.

| | U-Net | GAN | SDE |
|---|---|---|---|
| **Median** | 2.28 | -0.21 | 1.59 |
| **Absolute Median** | 14.6 | 11.8 | 11.5 |

Table 2: Median of bias change (% change in EODD/SER) by reconstruction approach across all datasets, tasks, and attributes by model. SDE and GAN show a slightly smaller bias shift than U-Net.

### 4.3. Bias Mitigation

While the impact of reconstruction on fairness was generally modest, applying mitigation strategies at the reconstruction stage could still reduce these effects or even improve the fairness of the underlying diagnostic models. We therefore tested two mitigation techniques inspired by classification literature but applied exclusively during reconstruction model training: sample reweighting and an equalized odds (EODD) constraint.

| | Sex | Age | Race |
|---|---|---|---|
| **Standard** | 24.1 | -1.88 | 3.30 |
| **Reweighted** | 10.6 | 0.03 | 1.05 |
| **EODD** | 7.56 | -2.01 | 0.52 |

Table 3: Median bias change (% change in EODD/SER) by mitigation strategy across all datasets, tasks, and models. Standard corresponds to the original results without mitigation applied.

Table 3 summarizes the bias changes for the mitigated models compared to the standard models. The summary is presented as an aggregation over pathologies and reconstruction model types, with results for each combination presented in Figures 4 and 5. Sex-related biases see the most substantial percentage improvements for both mitigation strategies. The EODD mitigation approach exhibited slightly lower median bias for each sensitive attribute than the reweighted sampling strategy (Table 3). For CheXpert the improved fairness for sex was most notable for the U-Net and SDE models, and less so for the GAN-based Pix2Pix model (Figure 4). For UCSF-PDGM segmentation, EODD reduced bias for most attributes and models, most strongly for U-Net (Figure 5). Classification fairness on UCSF-PDGM exhibits no consistent pattern, with fluctuations in both directions. Overall, while some fairness improvements are observed, the magnitudes are modest compared to the original bias (e.g., 16.5% median improvement for sex with EODD mitigation) and can depend on the pathology and sensitive attribute.

Fairness gains can incur performance trade-offs, but the trade-offs observed here are modest. Table 4 reports the mean change in PSNR and downstream task performance across reconstruction models when the mitigation strategies are applied. CheXpert deviations are below 1 % for PSNR and downstream AUROC. Downstream performance in UCSF-PDGM is also only moderately affected by the mitigation strategies, though PSNR shows

|  | Reweighted | | EODD | |
|---|---|---|---|---|
|  | Chex | UCSF | Chex | UCSF |
| **PSNR** | 0.54 | -0.75 | -0.64 | -7.28 |
| **Down.** | 0.07 | -1.97 | 0.02 | -2.94 |

Table 4: Mean change (%) in PSNR and downstream performance (AUROC/Dice) per dataset after each mitigation averaged over reconstruction models and tasks. Performance drops are modest, except for PSNR in UCSF-PDGM for EODD.

larger drops with EODD (see Figure 13 in the Appendix). Reweighting incurs the smallest penalties overall.

Additional results using EOP and $\Delta$Dice fairness metrics before and after mitigation are provided in the Appendix (Figures 14 and 15) and support the trends described above.

## 5. Discussion

We developed and applied an analysis framework that integrates reconstruction and prediction models to evaluate the effects of image reconstruction on downstream clinical performance and fairness, and investigate bias mitigation strategies at the reconstruction stage. Our analysis revealed several important insights, as summarized below.

**Stability of Downstream Performance:** Despite notable reductions in image quality, indicated by decreased PSNR at higher noise levels, downstream segmentation and classification performances remained robust to image reconstruction. This stability suggests that current diagnostic models are largely resilient to reconstruction-induced image degradations, at least for the studied tasks, which implies that minor reconstruction noise might not adversely impact clinical diagnostic accuracy. This finding may be surprising given that deep learning classification models are often thought to lack robustness, such as showing changes if the data are heterogeneous or noisy (Chuah et al., 2024). This suggests a nuanced interpretation of robustness, where models may be robust to certain transformations (e.g., reconstruction noise) but not others. Critically, these findings also have implications for the studied reconstruction models. Even if the downstream models were robust, we would expect that the performance of these models would drop if the reconstruction models removed the true underlying information necessary to perform the tasks. Instead, we observe largely stable downstream performance even as PSNR decreases, suggesting retainment of diagnostic features for the studied tasks. Nonetheless, there was a mild dependence on task difficulty, where more subtle pathologies (e.g. lung lesions) showed larger performance effects, highlighting future opportunities of applying our framework to other subtle tasks where the implications of AI-based reconstruction are currently unknown.

**Fairness Implications and Variability:** The aggregate effect of reconstruction on fairness was relatively modest, though certain pathologies and sensitive attributes showed significant shifts. These shifts varied in magnitude and direction, with a tendency toward increased bias, especially for patient sex. In most cases, the magnitude represented only a small fraction of the bias already present in the diagnostic models, though some would

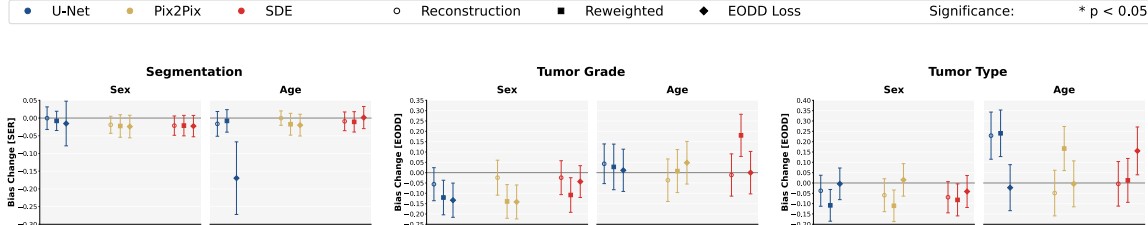

Figure 5: EODD and SER bias change pre- and post-mitigation compared to predictions on original images for UCSF-PDGM tasks. No consistent trends emerge for the classification tasks. Error bars represent standard deviation.

correspond to a ~5% difference in sensitivity/specificity between subgroups. Thus, reconstruction can contribute to bias in downstream tasks, but the overall bias appears to be largely driven by the downstream models themselves.

**Effectiveness of Mitigation Techniques:** Mitigation strategies reduced sex-related biases on CheXpert without measurable performance trade-offs in AUROC or PSNR (Figure 13 in the Appendix). However, similar mitigation strategies yielded inconsistent results on UCSF-PDGM, highlighting that their effectiveness is dataset-specific and dependent on the underlying task complexity and dataset characteristics.

**Sensitivity of Model Choice:** The SDE and GAN-based reconstruction approaches introduced lower additional bias overall compared to the standard U-Net, which may be counterintuitive given the generative nature of the SDE and GAN models. The U-Net also exhibited larger degradations in downstream performance when fine-tuned with the fairness mitigation strategies (Figure 13 in the Appendix). This sensitivity may arise from its inherently lower capacity than other methods, limiting simultaneous optimization of image fidelity and fairness constraints.

**Summary of Clinical Implications:** The robustness of downstream performance to AI-based image reconstruction is encouraging, particularly as these technologies are increasingly integrated into clinical practice. However, some performance drops were observed, especially for more subtle pathologies, highlighting the importance of rigorous evaluation and real-world monitoring. The potential for fairness shifts also necessitates active monitoring and reporting. This is especially important because model behavior can change as data distributions shift.

**Summary of Model Development Implications:** Developers of reconstruction models should prioritize downstream task and fairness evaluations alongside traditional pixel-level metrics, recognizing that reconstruction-induced biases, though subtle, can propagate through diagnostic workflows. This is especially the case for patient sex, where anatomical differences can be more prominent and may explain the larger effects observed for this attribute in our results. Bias mitigation strategies applied at the reconstruction stage may help improve fairness, but our results suggest that direct intervention at the classifier stage should be prioritized. Future research should explore multi-stage bias mitigation, integrat-

ing reconstruction and classification levels to achieve balanced fairness and performance outcomes.

**Limitations:** For comprehensiveness, we assessed multiple reconstruction models, downstream tasks, pathologies, and mitigation strategies, but this breadth necessarily creates challenges in data interpretation. As such, we have provided both summary level (e.g., Figure 3) and individual (e.g., Figure 4) results to enhance interpretability. Along with our studied datasets and tasks, it will be important in future work to apply our framework to additional datasets and clinical populations, including larger MRI datasets, to further probe generalization. Further generalization assessments for MRI should include additional sequences, directly modeling of k-space data rather than synthetic undersampling, and the inclusion of measurement noise rather than undersampling only. We note that our framework relies on the existence of adequate downstream models, which may not exist for every intended clinical task and dataset. However, the framework is agnostic to the type of downstream model, and could be applied to volumetric or temporal models, or could rely on self-supervised models before fine-tuning when labels are scarce. Additionally, while the algorithms used to create noisy images in this study simulate realistic acquisition degradations and are common approaches in the field (Feng, 2022; Gibson et al., 2023), they may not fully capture real-world variations.

## 6. Conclusion

The increasing clinical prevalence of AI-based reconstruction models creates a critical need for quantitative assessments of their potential downstream impact. We performed a scalable evaluation by using reconstruction and diagnostic AI models in tandem across multiple datasets, tasks, pathologies, and model types. We view our results as largely positive for the field – downstream performance was much more robust to reconstruction noise than image-level metrics, and the biases introduced by reconstruction were generally modest. However, some trends of increased bias were observed, especially for patient sex. Altogether, supported by these findings, we argue for the importance of monitoring downstream performance and fairness when using AI-based reconstruction models, and for continued work to mitigate emerging biases.

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

## Appendix A.

**Original**   **Pc: 100,000**   **Pc: 10,000**   **Pc: 3,000**

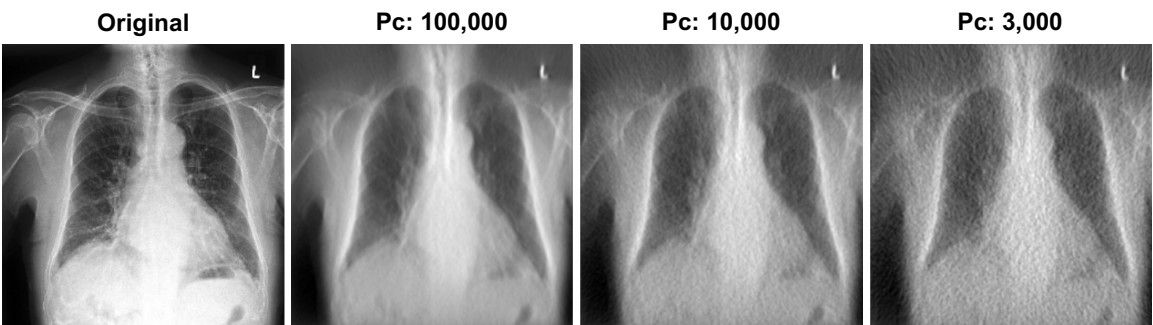

Figure 6: X-Ray images with photon count 100,000, 10,000, 3,000.

**Original**   **Acc: 4**   **Acc: 8**   **Acc: 16**

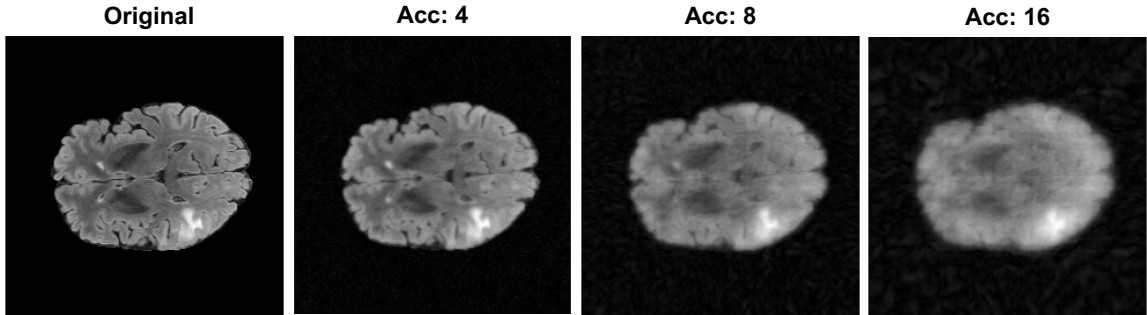

Figure 7: MRI images with acceleration 4, 8, 16.

**Diagnostic Hyperparameters.** The segmentation network was optimized with Adam (Kingma and Ba, 2014) using a learning rate of 0.001 and a batch size of 8 without data augmentation for 20 training epochs. The training loss consisted of Dice and L1, equally weighted at 0.5 each. The network used a sigmoid activation and a threshold of 0.5 was used at inference to compute the Dice performance. The model was trained on a per-slice level using all available MRI slices. At inference, Dice performance was computed using slices 60-130, as this range is representative of the regions where the ground truth masks appear and thus is more representative of performance. The Dice scores were computed separately for each slice, then averaged across slices per patient, followed by averaging across patients to compute final performance. For the UCSF-PDGM ResNet classifiers, we trained for 20 epochs with a learning rate of 0.0001 and a batch size of 16 without augmentation. Each task was treated as binary classification (subtype: glioblastoma vs not glioblastoma, grade: (II, III) vs IV) using binary cross entropy loss. All MRI slices were again used for training, followed by using slices 60-130 at inference. Prediction scores were generated separately for each slice, followed by computing a patient-level score as the median across slices to serve as input to patient-level AUROC calculations. The median across slices was

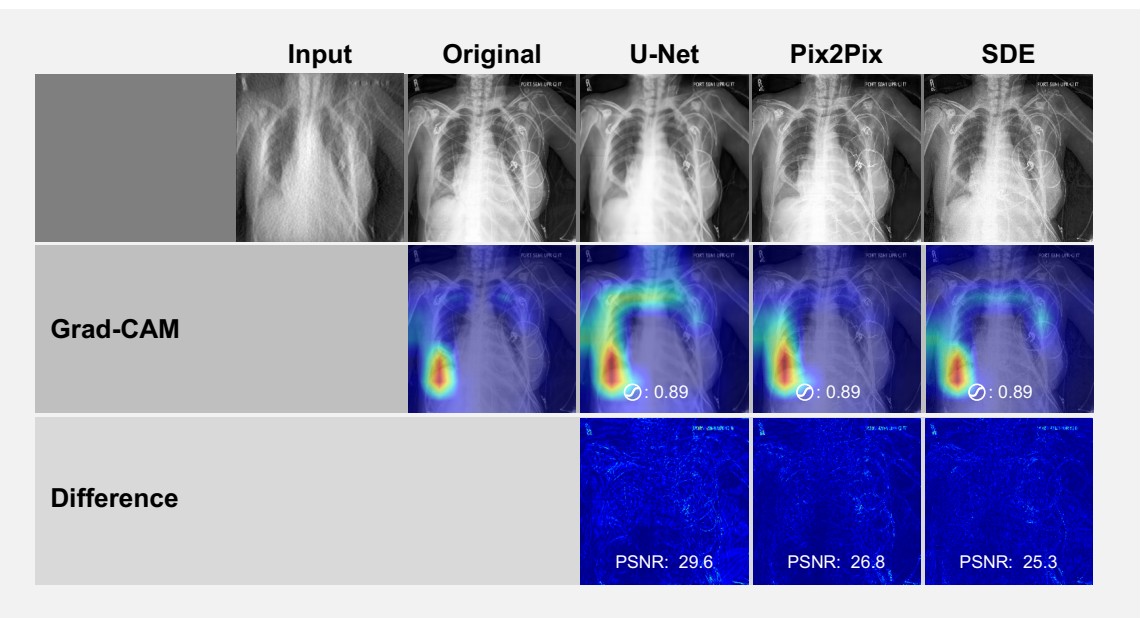

Figure 8: Reconstruction example from photon count 10,000 for the different models. Grad-CAM (Selvaraju et al., 2017) and logit score correspond to the lung lesion prediction of the pre-trained classifier, indicating similar predictions on the reconstructed images.

used to improve robustness to outliers. All UCSF-PDGM diagnostic models were trained using images pre-processed using min-max normalization to the 0-1 range and resized to 256x256. The CheXpert DenseNet classifier was trained using TorchXRayVision (Cohen et al., 2021). The default image preprocessing was used, with an input size of 224x224 pixels and normalization to a range of -1024 to 1024. The model was trained without data augmentation for 50 epochs using the Adam optimizer with a learning rate of 1e-3 and a weight decay of 1e-5.

**Reconstruction Hyperparameters.** No data augmentation was applied to any of the reconstruction pipelines. A U-Net was trained for 20 epochs on both UCSF-PDGM and CheXpert, using Adam with MSE loss, a learning rate of 0.001, and a batch size of 16. The GAN (Pix2Pix) was trained for 200 epochs on each dataset with Adam, a learning rate 0.0002, and a batch size of 32 to compensate for the smaller data volume. For the SDE model, we employed Adam with a learning rate of 0.0001, a cosine learning-rate schedule, and a batch size of 8; training ran for 40 epochs on CheXpert and 300 epochs on UCSF-PDGM. We note that the number of epochs varied between models because the different approaches take longer to converge (e.g., GANs are inherently less stable than a standard MSE loss), but in each case, the final weights were selected via validation loss monitoring, consistent with standard practice. During mitigation with the EODD-constraint, we employed $\tau = 0.5$ for the threshold, $T = 0.3$ for the temperature, and a momentum value of

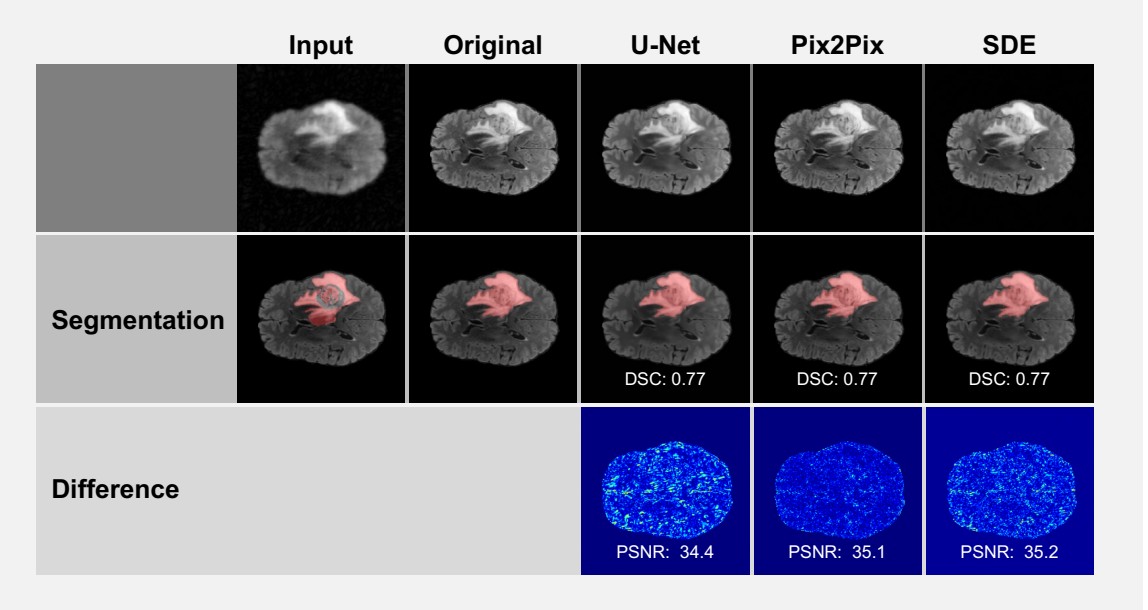

Figure 9: Reconstruction with corresponding segmentation and Dice score of an MRI image with acceleration 8 for the different models.

0.1 for the EMA. The remaining hyperparameters and architectural details were adopted unchanged from the original U-Net (Ronneberger et al., 2015), Pix2Pix (Isola et al., 2017), and SDE (Luo et al., 2023) publications. Image pre-processing consisted of min-max normalization to the 0-1 range and resizing to 256x256 for all reconstruction models.

The models were trained on a single NVIDIA A40 or A100 GPU. The SDE model was computationally most expensive and needed a maximum of 48 hours to train from scratch. For all models, the final weights were chosen based on performance on the validation split during training.

**MRI preprocessing and reconstruction details.** UCSF-PDGM provides reconstructed single-channel images (no multi-coil raw k-space data or complex-valued images). As a result, no coil combination, coil compression, or sensitivity map estimation was performed. To simulate undersampled MRI acquisitions, reconstructed images were retrospectively transformed to synthetic k-space using a discrete Fourier transform, implicitly assuming zero phase. Radial undersampling masks (Feng, 2022) were applied in k-space, and zero-filled reconstructions were obtained via inverse Fourier transform. The models were then trained in an image-to-image fashion to map the zero-filled images to the original reconstructed images. This pipeline was chosen to align with standard practice in MRI reconstruction studies when raw k-space is unavailable. The original UCSF-PDGM dataset was acquired using a 3.0 tesla scanner and a dedicated 8-channel head coil (Calabrese et al., 2022). Two gadolinium-based contrast agents were used across the cohort: gadobutrol at a dose of 0.1 mL/kg and gadoterate at a dose of 0.2 mL/kg.

**Proof of Proportionality.** When the protected attribute A takes more than two categories (e.g., multiple races, genders, or age groups), we compare all pairs $a_i, a_j$ of subgroups. Then, we take the maximum of the pairwise disparities in true positive and false positive rates:

$$
\begin{aligned}
EODD = \max_{1 \leq i < j \leq k} \Big[ & \big| P(\hat{Y} = 1 \mid Y = 1, A = a_i) \\
& - P(\hat{Y} = 1 \mid Y = 1, A = a_j) \big| \\
& + \big| P(\hat{Y} = 1 \mid Y = 0, A = a_i) \\
& - P(\hat{Y} = 1 \mid Y = 0, A = a_j) \big| \Big]
\end{aligned}
$$

Each pairwise comparison is handled exactly as in the binary case by treating $a_i, a_j$ as $0, 1$. Therefore, all the steps below—derived under a binary setup—apply pairwise to any two subgroups. Taking the maximum over these pairwise disparities then yields the multi-group measure.

This proof is based on the derivation by (Marcinkevics et al., 2022), and adjusted for EODD. EODD measures the disparity between subgroups in true positive rate (TPR) and false positive rate (FPR). In the binary case:

$$
\begin{aligned}
EODD = {} & P_{X,Y,A}(\hat{Y} = 1 | Y = 1, A = 1) \\
& - P_{X,Y|A}(\hat{Y} = 1 | Y = 1, A = 0) \\
& + P_{X,Y,A}(\hat{Y} = 1 | Y = 0, A = 1) \\
& - P_{X,Y,A}(\hat{Y} = 1 | Y = 0, A = 0)
\end{aligned}
$$

This can be expressed by the following proxy function.

$$
EODD = \frac{\sum_{i=1}^{n} f_\theta(x_i) a_i y_i}{\sum_{i=1}^{n} a_i y_i} \tag{1}
$$

$$
- \frac{\sum_{i=1}^{n} f_\theta(x_i)(1 - a_i) y_i}{\sum_{i=1}^{n} (1 - a_i) y_i} \tag{1}
$$

$$
+ \frac{\sum_{i=1}^{n} f_\theta(x_i) a_i (1 - y_i)}{\sum_{i=1}^{n} a_i (1 - y_i)} \tag{2}
$$

$$
- \frac{\sum_{i=1}^{n} f_\theta(x_i)(1 - a_i)(1 - y_i)}{\sum_{i=1}^{n} (1 - a_i)(1 - y_i)} \tag{2}
$$

To start, let's define the conditional covariance:

$$
\operatorname{cov}(A, X | Y = y) = \tag{3}
$$
$$
\mathbb{E}[(A - \mathbb{E}[A | Y = y])(X - \mathbb{E}[X | Y = y]) | Y = y]
$$
$$
= \mathbb{E}[AX | Y = y] - \mathbb{E}[A | Y = y]\mathbb{E}[X | Y = y] \tag{3}
$$

We can use the law of total covariance to prove the validity:

$$\text{cov}(A, X) = \mathbb{E}\Big[\text{cov}(A, X|Y)\Big] \tag{4}$$

$$+ \text{cov}\Big(\mathbb{E}[A|Y], \mathbb{E}[X|Y]\Big) \tag{4}$$

Expanding the first expectation term with (3):

$$\mathbb{E}[\text{cov}(A, X|Y)] = \mathbb{E}\Big[\mathbb{E}[AX|Y] - \mathbb{E}[A|Y]\mathbb{E}[X|Y]\Big]$$
$$= \mathbb{E}[AX] - \mathbb{E}[\mathbb{E}[A|Y]\mathbb{E}[X|Y]] \tag{5}$$

Expanding the second covariance term:

$$\text{cov}(\mathbb{E}[X|Z], \mathbb{E}[Y|Z]) = \mathbb{E}[\mathbb{E}[X|Z]\mathbb{E}[Y|Z]] \tag{6}$$
$$- \mathbb{E}[X]\mathbb{E}[Y] \tag{6}$$

Substituting (5) and (6) into (4):

$$\text{cov}(X, Y) = \mathbb{E}[XY] - \mathbb{E}[\mathbb{E}[X|Z]\mathbb{E}[Y|Z]]$$
$$+ \mathbb{E}[\mathbb{E}[X|Z]\mathbb{E}[Y|Z]] - \mathbb{E}[X]\mathbb{E}[Y]$$
$$= \mathbb{E}[XY] - \mathbb{E}[X]\mathbb{E}[Y]$$
$$= \text{cov}(X, Y)$$

We want to show that $\Delta_{OOD} \propto \widehat{\text{Cov}}(A, f_\theta(X)|Y = 1) + \widehat{\text{Cov}}(A, f_\theta(X)|Y = 0)$
Let $\sum_i a_i y_i = S_{AY}, \quad \sum_i a_i = S_A, \quad \sum_i y_i = S_Y$.

**Expanding EODD**:
Expanding (1):

$$\frac{\sum_{i=1}^{N} f_\theta(x_i) a_i y_i}{\sum_{i=1}^{N} a_i y_i} - \frac{\sum_{i=1}^{N} f_\theta(x_i)(1 - a_i) y_i}{\sum_{i=1}^{N} y_i(1 - a_i) y_i}$$

$$= \frac{1}{S_{AY}} \sum_{i=1}^{N} f_\theta(x_i) a_i y_i - \frac{1}{S_Y - S_A} \sum_{i=1}^{N} f_\theta(x_i)$$

$$+ \frac{1}{S_Y - S_{AY}} \sum_{i=1}^{N} f_\theta(x_i) a_i y_i$$

$$= \frac{S_Y}{S_{AY}(S_Y - S_{AY})} \sum_{i=1}^{N} f_\theta(x_i) y_i a_i$$

$$- \frac{1}{S_Y - S_{AY}} \sum_{i=1}^{N} f_\theta(x_i) y_i$$

Note that:

$$\widehat{\mathrm{Cov}}(A, f_\theta(X)|Y = 1)$$

$$= \frac{\sum_{i=1}^n f_\theta(x_i)a_i y_i}{\sum_{i=1}^n y_i}$$

$$- \frac{\sum_{i=1}^n a_i y_i}{\sum_{i=1}^n y_i} \frac{\sum_{i=1}^n f_\theta(x_i)y_i}{\sum_{i=1}^n y_i}$$

$$= \frac{1}{S_Y} \sum_{i=1}^n f_\theta(x_i)a_i y_i$$

$$- \frac{S_{AY}}{S_Y^2} \sum_{i=1}^n f_\theta(x_i)y_i.$$

Showing (5) $\propto \widehat{\mathrm{Cov}}(A, f_\theta(X)|Y = 1)$
with factor $\frac{S_Y^2}{S_{AY}(S_Y - S_{AY})}$, independent of $f_\theta$.

Expanding (2):

$$\frac{\sum_{i=1}^n f_\theta(x_i)a_i(1 - y_i)}{\sum_{i=1}^n a_i(1 - y_i)}$$

$$- \frac{\sum_{i=1}^n f_\theta(x_i)(1 - a_i)(1 - y_i)}{\sum_{i=1}^n (1 - a_i)(1 - y_i)}$$

$$= \frac{N - S_Y}{(N - S_Y - S_A + S_{AY})(S_A - S_{AY})} \sum_{i=1}^N f_\theta(x_i)a_i$$

$$- \frac{N - S_Y}{(N - S_Y - S_A + S_{AY})(S_A - S_{AY})} \sum_{i=1}^N f_\theta(x_i)a_i y_i$$

$$- \frac{1}{N - S_Y - S_A + S_{AY}} \sum_{i=1}^N f_\theta(x_i)y_i$$

$$- \frac{N}{N - S_Y - S_A + S_{AY}} \sum_{i=1}^N f_\theta(x_i)$$

Similarly:

$$
\begin{aligned}
\widehat{\mathrm{Cov}}&(A, f_0(X)|Y = 0) \\
&= \frac{\sum_{i=1}^{N} f_0(x_i)a_i(1 - y_i)}{\sum_{i=1}^{N}(1 - y_i)} \\
&\quad - \frac{\sum_{i=1}^{N} a_i(1 - y_i)}{\sum_{i=1}^{N}(1 - y_i)} \cdot \frac{\sum_{i=1}^{N} f_0(x_i)(1 - y_i)}{\sum_{i=1}^{N}(1 - y_i)} \\
&= \frac{1}{N - S_Y} \sum_{i=1}^{N} f_0(x_i)a_i \\
&\quad - \frac{N}{N - S_Y} \sum_{i=1}^{N} f_0(x_i)a_i y_i \\
&\quad\quad - \frac{S_A - S_{AY}}{(N - S_Y)^2} \sum_{i=1}^{N} f_0(x_i) \\
&\quad\quad - \frac{S_A \cdot S_{AY}}{(N - S_Y)^2} \sum_{i=1}^{N} f_0(x_i)y_i
\end{aligned}
$$

Showing $(6) \propto \widehat{\mathrm{Cov}}(A, f_\theta(X)|Y = 0)$  with factor  $\frac{(S_A - S_{AY})(N - S_Y - S_A + S_{AY})}{(N - S_Y)^2}$, independent of $f_\theta$.

Therefore, $EODD \propto \widehat{\mathrm{Cov}}(A, f_\theta(X)|Y = 1) + \widehat{\mathrm{Cov}}(A, f_\theta(X)|Y = 0)$.

|  | AI/AN | Asian | Black | NH/PI | Other | White |  |
|---|---|---|---|---|---|---|---|
| **Female,** $> 62$ | 54 | 1539 | 923 | 314 | 2518 | 6456 | 11804 |
| **Female,** $\leq 62$ | 39 | 1739 | 608 | 136 | 1710 | 9500 | 13732 |
| **Male,** $> 62$ | 56 | 1734 | 1023 | 240 | 3553 | 8984 | 15590 |
| **Male,** $\leq 62$ | 27 | 1924 | 539 | 171 | 1853 | 11170 | 15684 |
|  | 176 | 6936 | 3093 | 861 | 9634 | 36110 | 56810 |

Table 5: Patient-wise groups used for analysis based on sex, age, and race for the CheXpert dataset. Unequally distributed with very few samples for American Indian or Alaska Native (AI/AN) and Native Hawaiian or Other Pacific Islander (NH/PI).

|  | Male | Female |  |
|---|---|---|---|
| $\leq 58$ | 155 | 92 | 147 |
| $> 58$ | 144 | 110 | 254 |
|  | 299 | 202 | 501 |

Table 6: Patient distribution by sex and age for the UCSF-PDGM dataset. Patients under 58 and females represent minority groups.

**Additional Fairness Results**

In addition to Equalized Odds and Skewed Error Ratio in the main text, we investigate two additional bias metrics:

**Equality of Opportunity (EOP):**

$$P(\hat{Y} = 1 \mid Y = 1, A = 0)$$
$$= P(\hat{Y} = 1 \mid Y = 1, A = 1).$$

We report the worst case Equality of Opportunity (Hardt et al., 2016) difference between groups

$$max_{i,j} | P(\hat{Y} = 1 \mid Y = 1, A = i)$$
$$- P(\hat{Y} = 1 \mid Y = 1, A = j)|,$$
$$\forall \quad A \in \mathcal{A}.$$

EOP is a relaxation of EODD, requiring fairness only concerning the positive class $(Y = 1)$.

**$\Delta$Dice:** Given the limited availability of dedicated segmentation fairness metrics, we also compute:
$$\Delta\text{Dice} = \max_{i,j} \left| \text{Dice}_{A_i} - \text{Dice}_{A_j} \right|, A \in \mathcal{A}$$

which represents the maximum difference in Dice across all protected subgroups $\mathcal{A}$.

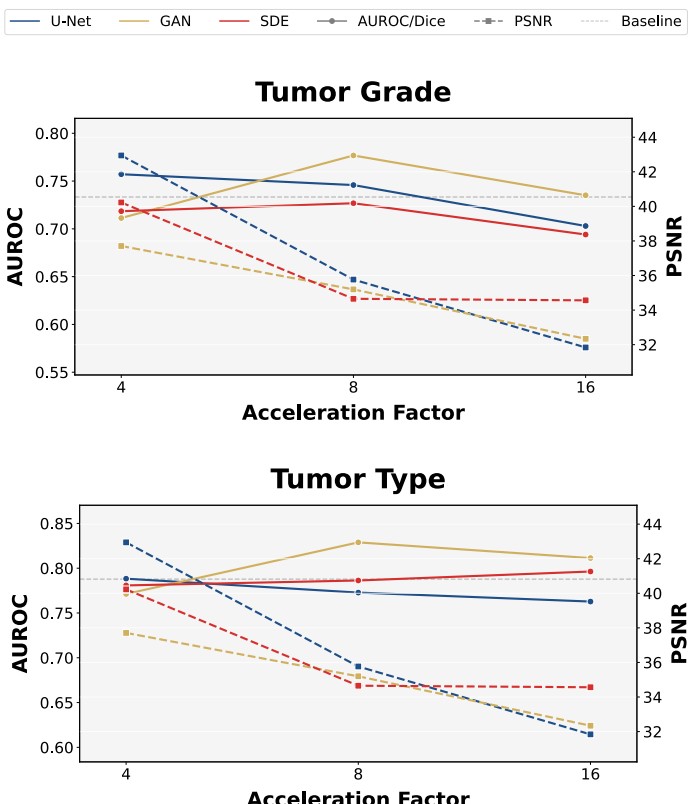

Figure 10: Tumor Type and Tumor Grade and PSNR values for different noise levels on UCSF-PDGM. The image quality and diagnostic performance axes are on a similar percentage scale. Task performance metrics show high stability across models and noise conditions, while PSNR drops with increasing noise.

Plots containing the results of these additional evaluations can be found in Figure 14 and 15.

Additionally, Figures 16 and 17 contain results using different race subgroups for CheX-pert. Our original evaluations considered each of the original subgroups listed within the dataset (Table 5) when computing the fairness metrics. Given the small counts for the American Indian or Alaska Native and Native Hawaiian or Other Pacific Islander subgroups, leading to large error bars, we also computed these metrics when including these subgroups within the Other subgroup.

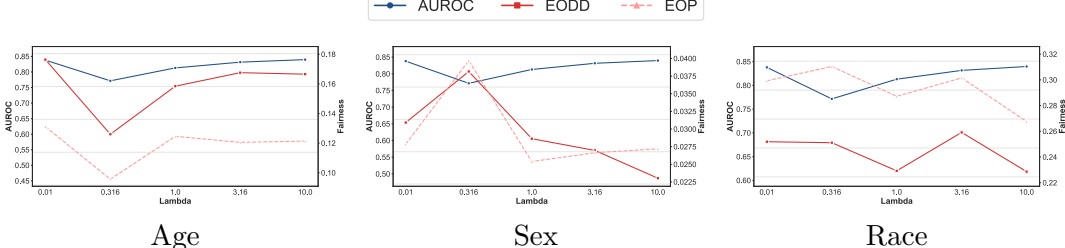

Age             Sex             Race

Figure 11: Influence of fairness weighting parameter ($\lambda_{\text{fair}}$) on classifier AUROC performance and fairness metrics for the Equalized Odds (EODD) mitigation constraint, evaluated with U-Net on the CheXpert dataset. There is minor sensitivity of AUROC to lambda; fairness metrics show greater variance but minimal substantial improvement with increased $\lambda$.

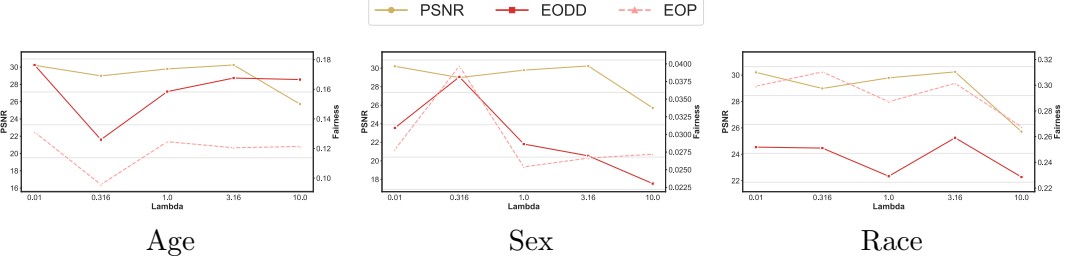

Age             Sex             Race

Figure 12: Impact of $\lambda_{\text{fair}}$ on reconstruction quality (PSNR) compared to fairness for the EODD constraint mitigation. PSNR remains stable across lambda variations, while fairness shows slight variation without substantial improvement.

| Photon Count | | Metrics | Baseline | U-Net | GAN | SDE |
|---|---|---|---|---|---|---|
| | | Atelectasis | 0.87 | 0.87 | 0.86 | 0.87 |
| | | Cardiomegaly | 0.91 | 0.91 | 0.91 | 0.91 |
| | | Consolidation | 0.91 | 0.91 | 0.91 | 0.91 |
| | | Edema | 0.90 | 0.90 | 0.90 | 0.90 |
| | | EC | 0.79 | 0.78 | 0.78 | 0.79 |
| | | Fracture | 0.76 | 0.75 | 0.75 | 0.76 |
| | AUROC | Lung Lesion | 0.80 | 0.79 | 0.79 | 0.79 |
| 100,000 | | Lung Opacity | 0.88 | 0.88 | 0.88 | 0.88 |
| | | Pleural Effusion | 0.93 | 0.92 | 0.92 | 0.92 |
| | | Pleural Other | 0.83 | 0.82 | 0.81 | 0.82 |
| | | Pneumonia | 0.83 | 0.83 | 0.83 | 0.83 |
| | | Pneumothorax | 0.77 | 0.75 | 0.76 | 0.77 |
| | | Average | 0.85 | 0.84 | 0.84 | 0.85 |
| | PSNR | | | 31.60 | 30.16 | 29.98 |
| | LPIPS | | | 0.13 | 0.08 | 0.08 |
| | | Atelectasis | 0.87 | 0.87 | 0.86 | 0.87 |
| | | Cardiomegaly | 0.91 | 0.90 | 0.90 | 0.91 |
| | | Consolidation | 0.91 | 0.91 | 0.90 | 0.91 |
| | | Edema | 0.90 | 0.89 | 0.89 | 0.90 |
| | | EC | 0.79 | 0.78 | 0.78 | 0.78 |
| | | Fracture | 0.76 | 0.75 | 0.74 | 0.75 |
| | AUROC | Lung Lesion | 0.80 | 0.78 | 0.78 | 0.79 |
| 10,000 | | Lung Opacity | 0.88 | 0.88 | 0.87 | 0.88 |
| | | Pleural Effusion | 0.93 | 0.92 | 0.91 | 0.92 |
| | | Pleural Other | 0.83 | 0.81 | 0.80 | 0.82 |
| | | Pneumonia | 0.83 | 0.82 | 0.82 | 0.82 |
| | | Pneumothorax | 0.77 | 0.75 | 0.75 | 0.77 |
| | | Average | 0.85 | 0.84 | 0.83 | 0.84 |
| | PSNR | | | 30.52 | 28.62 | 27.12 |
| | LPIPS | | | 0.19 | 0.11 | 0.15 |
| | | Atelectasis | 0.87 | 0.86 | 0.85 | 0.86 |
| | | Cardiomegaly | 0.91 | 0.90 | 0.90 | 0.91 |
| | | Consolidation | 0.91 | 0.91 | 0.90 | 0.90 |
| | | Edema | 0.90 | 0.89 | 0.89 | 0.89 |
| | | EC | 0.79 | 0.78 | 0.78 | 0.78 |
| | | Fracture | 0.76 | 0.74 | 0.73 | 0.75 |
| | AUROC | Lung Lesion | 0.80 | 0.77 | 0.77 | 0.78 |
| 3000 | | Lung Opacity | 0.88 | 0.87 | 0.87 | 0.87 |
| | | Pleural Effusion | 0.93 | 0.91 | 0.91 | 0.92 |
| | | Pleural Other | 0.83 | 0.80 | 0.78 | 0.81 |
| | | Pneumonia | 0.83 | 0.82 | 0.80 | 0.82 |
| | | Pneumothorax | 0.77 | 0.74 | 0.74 | 0.77 |
| | | Average | 0.85 | 0.83 | 0.83 | 0.84 |
| | PSNR | | | 28.89 | 27.36 | 26.83 |
| | LPIPS | | | 0.22 | 0.14 | 0.15 |

Table 7: CheXpert performance across reconstruction models and photon counts. Pathologies with lower baseline AUROC (e.g., fracture, pneumothorax, lung lesion) experience greater performance drops under noise compared to more easily detectable conditions (e.g., effusion, cardiomegaly). Baseline corresponds to original images.

| Acceleration | Metrics | | Baseline | U-Net | GAN | SDE |
|---|---|---|---|---|---|---|
| 4 | **AUROC** | **Tumor Type** | 0.79 | 0.79 | 0.77 | 0.78 |
| | | **Tumor Grade** | 0.73 | 0.76 | 0.71 | 0.72 |
| | **Dice** | | 0.72 | 0.72 | 0.71 | 0.72 |
| | **PSNR** | | | 42.94 | 37.71 | 40.23 |
| | **LPIPS** | | | 0.01 | 0.02 | 0.00 |
| 8 | **AUROC** | **Tumor Type** | 0.79 | 0.77 | 0.83 | 0.79 |
| | | **Tumor Grade** | 0.73 | 0.75 | 0.78 | 0.73 |
| | **Dice** | | 0.72 | 0.70 | 0.71 | 0.71 |
| | **PSNR** | | | 35.77 | 35.20 | 34.65 |
| | **LPIPS** | | | 0.03 | 0.02 | 0.02 |
| 16 | **AUROC** | **Tumor Type** | 0.79 | 0.76 | 0.81 | 0.80 |
| | | **Tumor Grade** | 0.73 | 0.70 | 0.74 | 0.69 |
| | **Dice** | | 0.72 | 0.67 | 0.70 | 0.71 |
| | **PSNR** | | | 31.84 | 32.34 | 34.56 |
| | **LPIPS** | | | 0.06 | 0.04 | 0.02 |

Table 8: Performance metrics for UCSF-PDGM across reconstruction models and noise levels. While PSNR varies with noise and model, downstream segmentation and classification metrics remain relatively stable, indicating robust task performance across conditions.

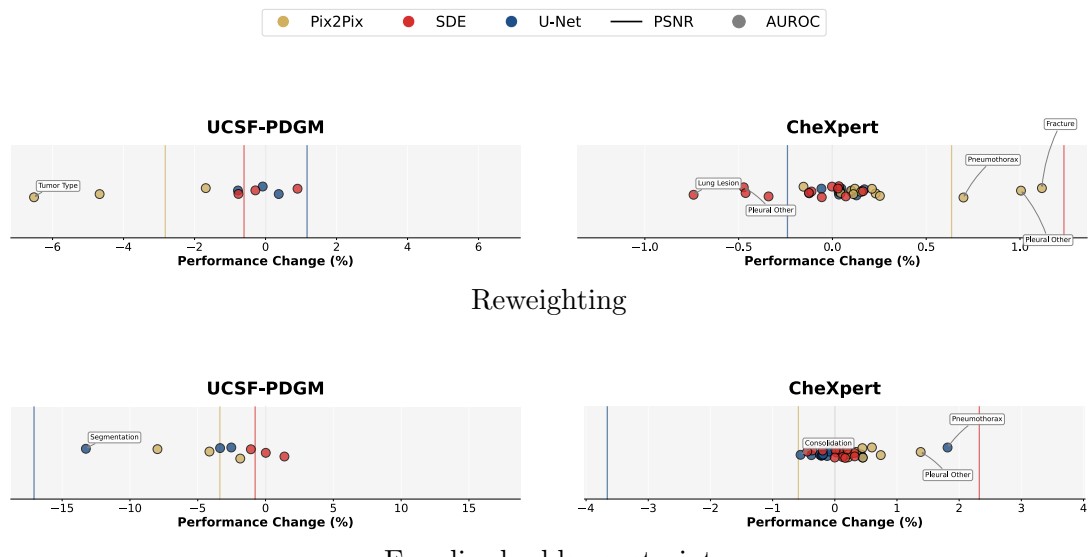

Reweighting

Equalized odds constraint

Figure 13: Change in prediction performance after applying bias mitigation techniques. Each row compares two datasets for a given method: (a) Reweighted sampling, (b) Equalized odds constraint. UCSF-PDGM experiences more performance degradation. However, all techniques show good stability in task performance, with few outliers in the UCSF-PDGM dataset.

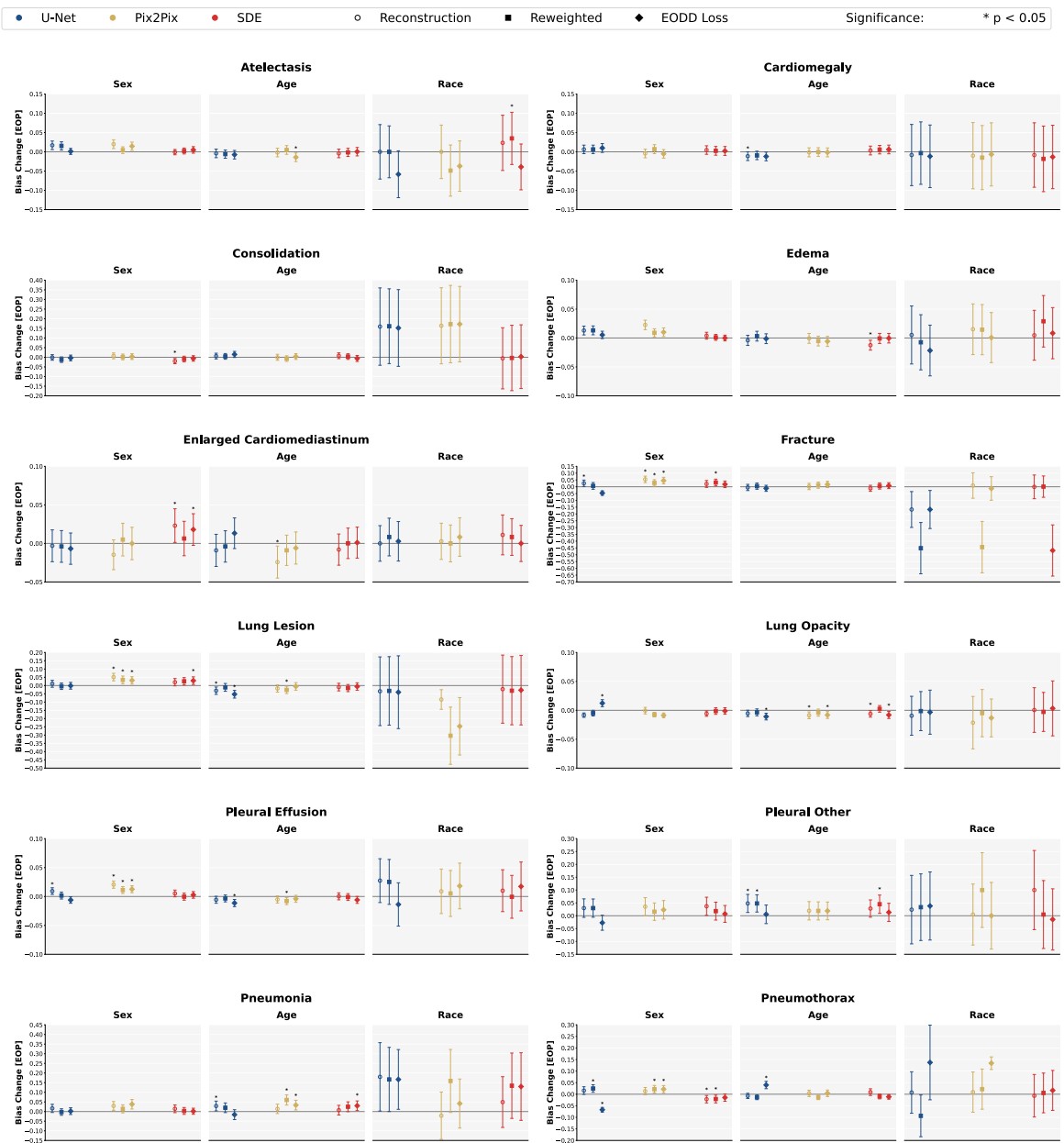

Figure 14: Equality of opportunity (EOP) bias change pre- and post-mitigation compared to predictions on original images for CheXpert classification. Pre-mitigation, bias tends to increase slightly for sex; race exhibits high variance. Bias tends to decline slightly post-mitigation.

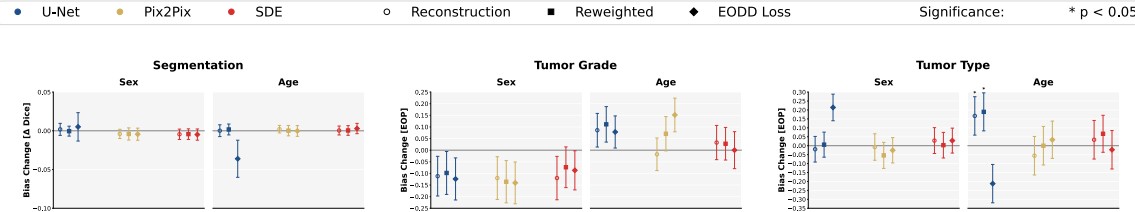

Figure 15: Equality of opportunity (EOP) and $\Delta$ Dice bias change compared to predictions on original images pre- and post-mitigation for UCSF-PDGM classification and segmentation.

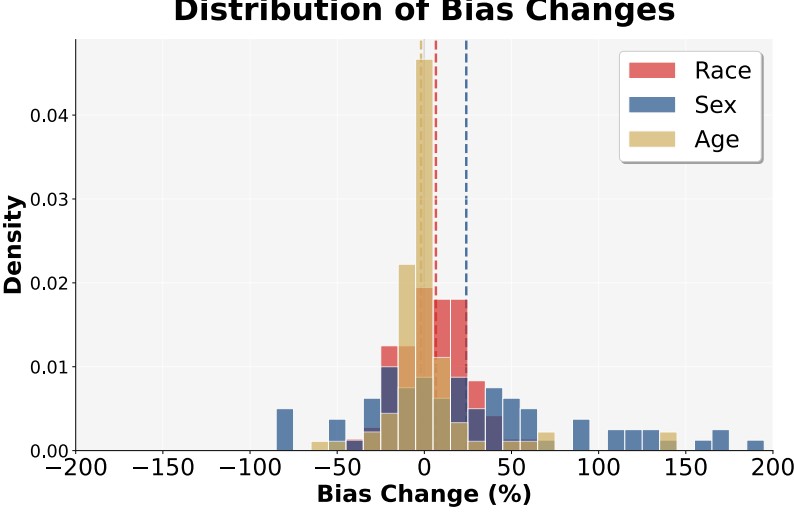

Figure 16: Distribution of bias changes when using alternative race subgroups for CheXpert calculations.

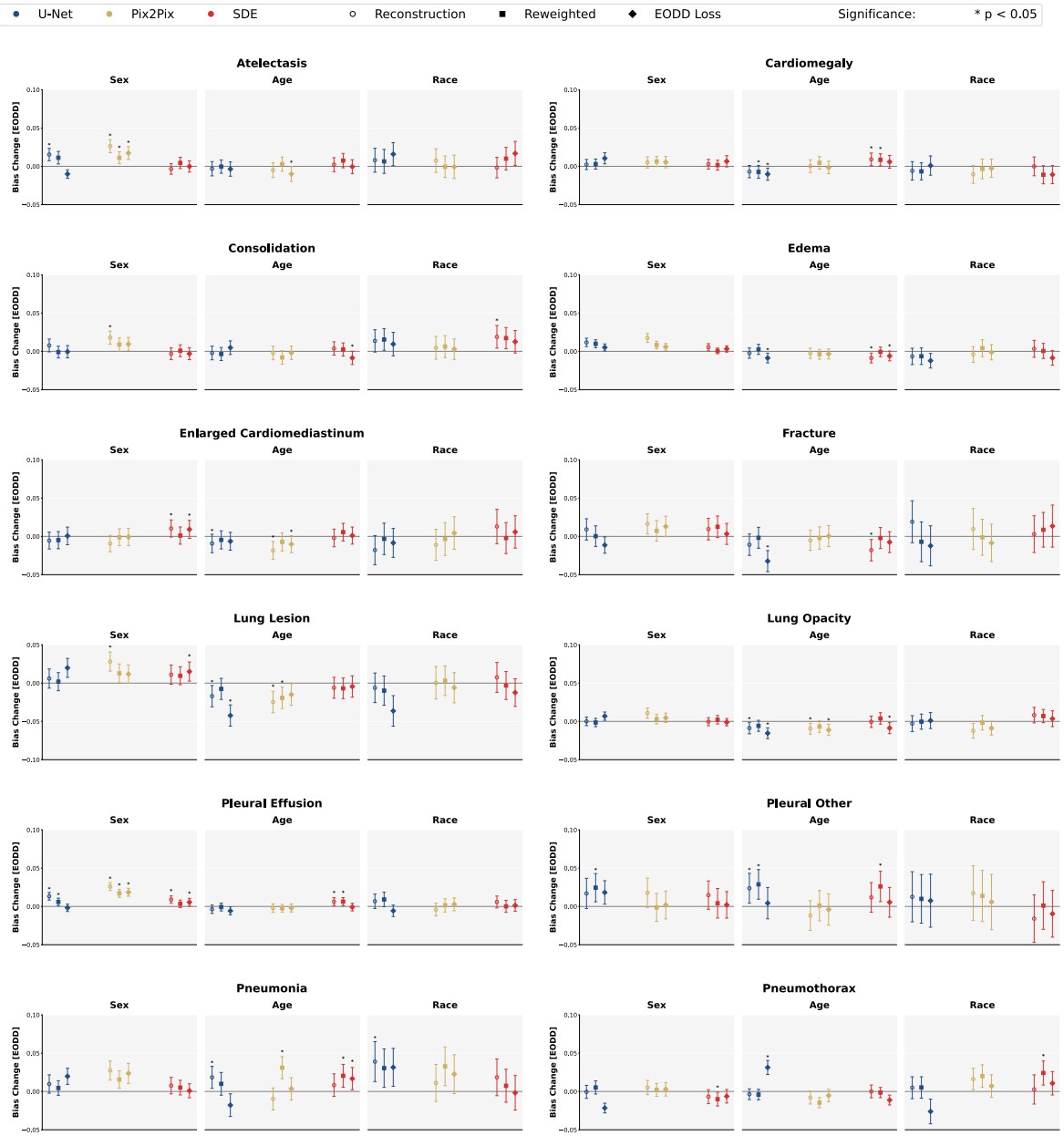

Figure 17: Equalized odds bias change pre- and post-mitigation compared to predictions on original images for CheXpert when using alternative race subgroups.

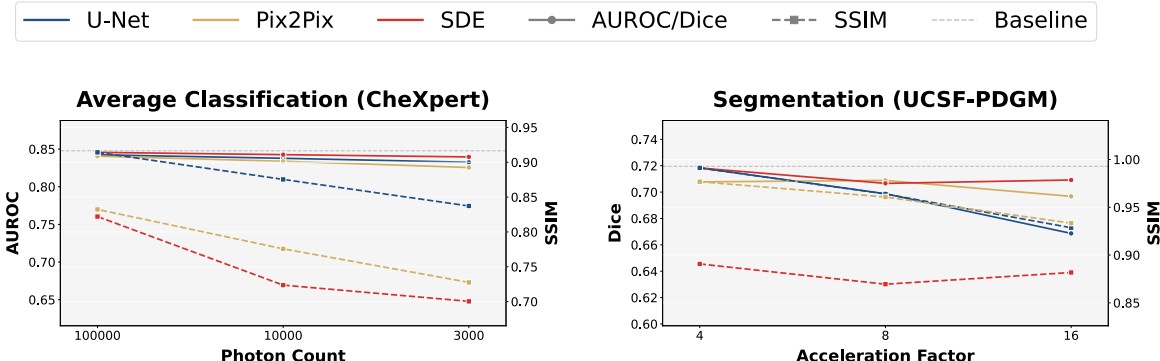

Figure 18: Downstream performance and SSIM at varying noise levels. Axes for SSIM and task performance are scaled to comparable percentage ranges. Baseline indicates performance on original images.

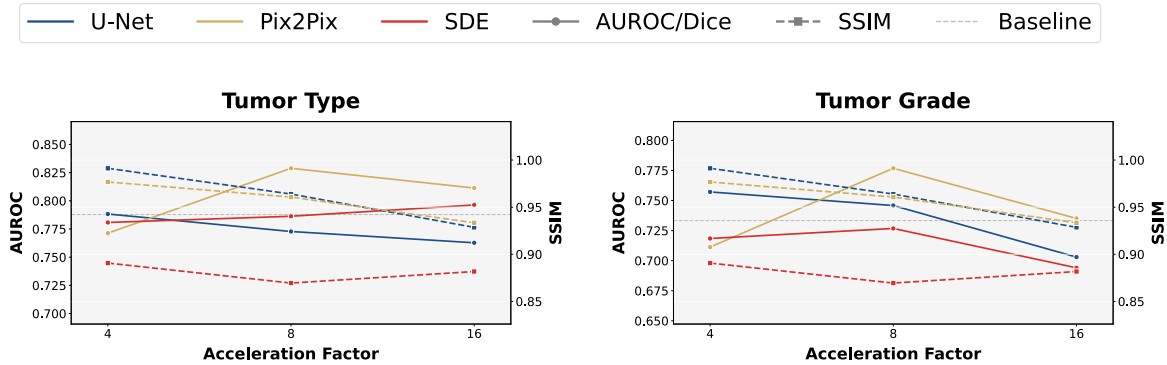

Figure 19: Downstream performance and SSIM at varying noise levels on classification tasks in UCSF-PDGM. Axes for SSIM and task performance are scaled to comparable percentage ranges. Baseline indicates performance on original images.

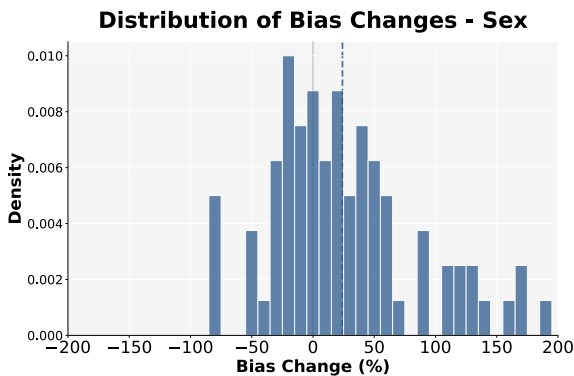

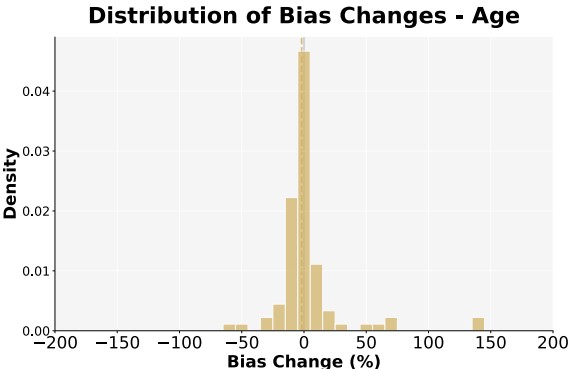

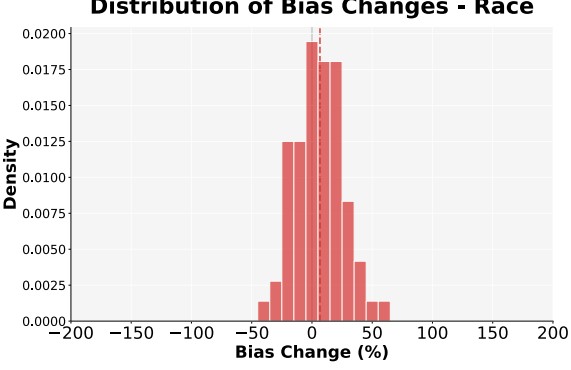

Figure 20: Distribution of bias changes separated by sensitive attribute.

