# OpenReview forum: "Evaluating the Impact of Medical Image Reconstruction on Downstream AI Fairness and Performance"
_MIDL.io/2026/Validation_Papers — MIDL 2026 - Validation Papers Poster_

### Official Review · Reviewer_uBfZ · 2026-01-09

**Confidence:** 4
**Preliminary Rating:** 4
**Final Rating:** 5

**Summary:**

The authors present a unified evaluation framework that links image reconstruction from noisy X-rays and undersampled MRI measurements with downstream clinical task performance. They evaluate two imaging modalities (MRI, X-ray) across large datasets/cohorts, considering two diagnostic tasks (classification, segmentation) and three reconstruction approaches (U-Net MSE–based, diffusion-based, GAN-based training mechanisms). Their results indicate that, as noise increases, downstream task performance remains relatively stable even when standard reconstruction quality metrics degrade. Overall, the paper advocates for a more holistic assessment of AI-based reconstruction systems by emphasizing clinically relevant downstream task performance alongside pixel-level image reconstruction quality.

**Strengths:**

- **Clinical Relevance**: The paper presents a unified evaluation framework that jointly assesses reconstruction quality and downstream diagnostic performance, which supports more clinically meaningful evaluation than reconstruction metrics alone. In addition, the work thoughtfully connects the effect of reconstruction to fairness/robustness in downstream tasks (e.g., sex/age subgroup performance), which is a novel perspective that is not commonly addressed in a single end-to-end analysis.
- **Comprehensive Evaluation**: The experimental validation is extensive, covering two imaging domains with large datasets (MRI: 501 patients; X-ray: 224,316 images from 65,240 patients) and two diagnostic task types (classification and segmentation), with multiple reconstruction approaches (U-Net–based, diffusion-based, and adversarial/GAN approaches).
- **Reproducibility support**: The authors provide strong reproducibility support, e.g., code and data availability, plus well-documented hyperparameters in Appendix.
- **Data demographics**: The authors report cohort variability (data demographics) in Table 6.
- **Presentation**: Overall, the paper is well-written and -presented, with strong supporting figures/tables and comprehensive appendices (pipeline overview, data characteristics, reconstruction/visual examples, and training parameters).

**Weaknesses:**

- Given that downstream task performance appears largely stable even as reconstruction metrics degrade (and reconstruction-related bias has limited downstream impact in most subgroups), it would be helpful to clarify the broader clinical utility and future directions of the proposed development, and to more explicitly define its intended scope. Given the observed stability, please explain when and why reconstruction would be expected to meaningfully affect downstream performance, and what scenarios would most benefit from the proposed bias-mitigation components.
- Since the framework relies on supervised training for reconstruction and downstream tasks, it would be helpful to discuss how it could be applied when labels are scarce (e.g., via self-/weakly-supervised diagnostic models), and to broaden the reconstruction literature review accordingly by citing relevant deep-learning reconstruction work, including recent unsupervised approaches e.g.,  [1], [2].
- If all data comes from a single acquisition protocol (e.g., FLAIR for MRI), generalization to different protocols/acquisition settings may be limited. If present, please describe any acquisition/protocol variation to better support generalizability.
- In addition to the limitations of dataset size/variety and simulation realism, please discuss framework-level limitations, such as applicability to dynamic/time-resolved imaging (e.g., motion in cardiac/lung MRI [1]) and what modifications would be required to extend the framework beyond static single-slice settings.

[1] Chen, Chong, et al. "A multi-dynamic low-rank deep image prior (ML-DIP) for 3D real-time cardiovascular MRI." Journal of Cardiovascular Magnetic Resonance (2025): 102015.

[2] Sultan, Muhammad Ahmad, et al. "An unsupervised method for MRI recovery: deep image prior with structured sparsity." Magnetic Resonance Materials in Physics, Biology and Medicine (2025): 1-13.

**Detailed Comments:**

- The MRI preprocessing and reconstruction setup needs more detail. Please clarify whether the MRI experiments use multi-coil raw kspace data; if so, how coils are combined/compressed, how sensitivity maps are estimated, and whether the reconstruction enforces data consistency in k-space or is purely image-to-image. Also, please summarize key MRI acquisition settings (e.g., field strength, sequence/contrast).
- How are the network inputs initialized for MRI and X-ray experiments? For MRI, is the input a zero-filled reconstruction from undersampled measurements?
- For realism in MRI simulation, did you model measurement noise (e.g., add white Gaussian noise) in addition to undersampling artifacts? If not, please discuss why it is not needed or how this may affect conclusions.
- If you used complex-valued MRI images, please clarify the representation used in the network (e.g., real/imaginary as two channels) and how reconstruction metrics are computed (PSNR/SSIM on magnitude vs complex images).
- Please report standard deviation across subjects/datasets in Table 1 along with mean.

**Justification Of Final Rating:**

The authors’ rebuttal and revised manuscript address my main concerns by substantially improving clarity around the scope and clinical implications of the observed downstream stability, expanding the discussion of when reconstruction may matter more, and strengthening the limitations and future directions. Overall, the paper provides a clinically relevant, comprehensive evaluation with strong reproducibility support.

**Justification Of The Preliminary Rating:**

The paper has sound validation-related strengths such as clear clinical relevance, comprehensive evaluation and reproducibility support. However, it lacks sufficient literation review, limitation discussion, and clear future directions.

**Questions To Address In The Rebuttal:**

Address weaknesses:
- Given that downstream task performance appears largely stable even as reconstruction metrics degrade (and reconstruction-related bias has limited downstream impact in most subgroups), it would be helpful to clarify the broader clinical utility and future directions of the proposed development, and to more explicitly define its intended scope.
- Given the observed stability, please explain when and why reconstruction would be expected to meaningfully affect downstream performance, and what scenarios would most benefit from the proposed bias-mitigation components.
- Since the framework relies on supervised training for reconstruction and downstream tasks, it would be helpful to discuss how it could be applied when labels are scarce (e.g., via self-/weakly-supervised diagnostic models), and to broaden the reconstruction literature review accordingly by citing relevant deep-learning reconstruction work, including recent unsupervised approaches e.g., [1], [2].
- If all data comes from a single acquisition protocol (e.g., FLAIR for MRI), generalization to different protocols/acquisition settings may be limited. If present, please describe any acquisition/protocol variation to better support generalizability.
- In addition to the limitations of dataset size/variety and simulation realism, please discuss framework-level limitations, such as applicability to dynamic/time-resolved imaging (e.g., motion in cardiac/lung MRI [1]) and what modifications would be required to extend the framework beyond static single-slice settings.

---

### Official Review · Reviewer_74nt · 2026-01-10

**Confidence:** 4
**Preliminary Rating:** 4
**Final Rating:** 5

**Summary:**

The authors evaluated the impact of three medical image reconstruction methods (U-Net, GAN, and diffusion) on downstream fairness and performance in the context of downstream tasks such as classification and segmentation. The authors found that diagnostic tools still work well even if the reconstructed images have lower quality scores, but the reconstruction process can add small biases (especially related to sex) that are hard to fix.

**Strengths:**

The experiments are carefully designed. The selection of reconstruction methods that trace the evolution of the field: from established baselines (U-Net) and adversarial approaches (GANs) to state-of-the-art diffusion models.

**Weaknesses:**

The diffusion model was not trained in the standard way. First, diffusion models usually require a huge amount of data to learn effectively, but this study used a very small dataset of 350 MRI scans (501 with 70/10/20 train/validation/test split). Second, instead of checking the actual visual quality of the generated images (using metrics like FID) to decide when to stop training, the authors selected the model based on the validation loss.

**Detailed Comments:**

Labels overlap with other labels and data points in Figure 15.
On page 30, the figure label overlaps with the page number.
Some figures are too small to see, such as Figure 4.
In Figure 3, the bars of different colors are blended together and difficult to distinguish. Please consider making the figure clearer for readers.

**Justification Of Final Rating:**

The authors addressed my concerns. They clarified diffusion training (including a large CheXpert dataset), showed results were consistent across methods, and fixed the figure issues. They also noted the need for larger MRI validation.

**Justification Of The Preliminary Rating:**

The authors made a significant effort in the experimental design, particularly in choosing a comprehensive range of models for evaluation that spans from early baselines (U-Net) and intermediate GANs to currently popular diffusion models. However, a notable weakness is the training of the diffusion model, which deviated from standard practices by relying on a small dataset and validation loss monitoring rather than the conventional training protocols found in the majority of literature.

**Questions To Address In The Rebuttal:**

For figure 9, I see less errors in the error map for Pix2Pix method but the PSNR of this method is 35.1 which is lower than SDE method. Can you explain why?

Diffusion models typically require large-scale datasets for training. Do you consider the size of the MRI dataset used in this paper sufficient? Furthermore, while diffusion models often use sampling metrics like FID to determine when to stop training, this study selected final weights based on validation loss monitoring. Do you believe these two factors: dataset size and the checkpoint selection criterion, might have limited the diffusion model's performance?

---

### Official Review · Reviewer_jJEv · 2026-01-10

**Confidence:** 3
**Preliminary Rating:** 3
**Final Rating:** 4

**Summary:**

This paper studies how AI-based image reconstruction affects downstream clinical tasks (classification and segmentation) and fairness on both MRI and X-ray datasets. It proposes an end-to-end evaluation framework that simulates acquisition degradation, applies reconstruction models (U-Net, GAN, and diffusion), and measures diagnostic performance and fairness, with optional bias mitigation strategies applied at the reconstruction stage. The downstream baseline models are trained on the original images and then evaluated on reconstructed images across increasing noise levels. Results suggest that downstream classification/segmentation performance is relatively robust to the tested noise levels, even as PSNR decreases. The paper also reports how reconstruction can shift subgroup disparities across sex, age, and race, and evaluates several mitigation methods.

**Strengths:**

(1) This paper provides a novel perspective on evaluating how reconstruction models perform on downstream tasks and provides the quantitative results of this.

(2) This paper provides comprehensive experiments on the reconstruction fairness,

**Weaknesses:**

(1) Generalization and practicality. The proposed evaluation relies on a strong downstream classification/segmentation (CLS/SEG) model. In common medical out-of-domain settings, such a model may not exist or may not transfer, so users would need to find a reliable pretrained model or train their own if they want to test their reconstruction models' performance on downstream tasks. This dependency can make the evaluation difficult to apply in practice.

(2) Although downstream CLS/SEG performance is stable across noise levels, its relevance for assessing reconstruction quality is unclear. High AUROC/Dice may simply reflect downstream-model robustness and can remain high even when PSNR drops sharply, which means the reconstruction quality is decreasing. Therefore, it is unclear why this evaluation is needed: it does not reflect reconstruction fidelity, and it only confirms the downstream model is robust, but does not provide more insights beyond it.

(3) Lack of metrics. There are other commonly used metrics to evaluate image reconstruction, such as SSIM mentioned in the related work.

**Detailed Comments:**

All my concerns are in the weakness section.

**Justification Of Final Rating:**

Thanks for the answers, and the authors have addressed all of my concerns. The idea is novel, and I believe this paper brings a practical evaluation tool as the pretrained models are available for some tasks. The explanation of PSNR and CLS scores is reasonable and convincing. Therefore, I raise my final score.

**Justification Of The Preliminary Rating:**

This paper provides a novel perspective on evaluating how reconstruction models perform on downstream tasks, and provides the quantitative results of this. It also does comprehensive experiments on the fairness. However, the practical usage and whether this evaluation can really reflect reconstruction quality are unclear.

**Questions To Address In The Rebuttal:**

(1) It would be helpful to discuss the generalization ability of this evaluation.

(2) It would strengthen the paper if the authors clarified and supported use cases where downstream task performance is the primary objective and reconstruction fidelity can be deprioritized. Otherwise, if the main takeaway of this part is that the chosen CLS/SEG models are robust to increasing noise, the practical contribution of this evaluation is limited.

(3) Add SSIM to figure 2 (or maybe a table to compare the SSIM under different noise levels).

I'm open to increasing my score if the authors address these questions.

---

### Author Rebuttal · Authors · 2026-01-25

**Rebuttal:**

We thank the Reviewers for their constructive comments and positive feedback, which we have incorporated into the attached updated manuscript. Changes in the manuscript consist of the following:
1) Text updates as detailed in our responses below. All text updates are bolded in red font.
2) Error bars added to Table 1.
3) Figure/table formatting: Removal of overlap between legend text and page number for Table 7. Removal of overlapping text labels in Figure 15. Addition of version of Figure 3 without overlapping bars (Figure 22).
4) Addition of SSIM results (Figures 20 and 21).

**Supporting Material:**

/attachment/c771293d042aec80890c3655e13cff6de714aa6f.pdf

---

### Meta-Review · Area_Chair_q34k · 2026-02-06

**Recommendation:** Accept (Oral)
**Confidence:** 5

**Metareview:**

This paper evaluates how modern medical image reconstruction (U-Net, GAN, diffusion) impacts downstream AI performance and fairness, across both X-ray and MRI settings and across classification/segmentation tasks. Reviewers agreed the experimental design is strong and the scope is unusually comprehensive for a validation paper, and the main message is important: standard reconstruction metrics (e.g., PSNR/SSIM) do not reliably predict downstream behavior, while subgroup fairness can shift in less predictable ways.

The initial concerns were mainly about (i) practical reliance on downstream models, (ii) whether stable downstream AUROC/Dice simply reflects robustness rather than meaningful reconstruction evaluation, and (iii) missing/limited reconstruction metrics and implementation details (including diffusion training and MRI reconstruction assumptions). The rebuttal and revision addressed these points well: SSIM analyses and error bars were added, diffusion training and data scale were clarified (including large-scale X-ray experiments), and the MRI reconstruction setup and limitations were explained more transparently. The discussion was also strengthened around when reconstruction is expected to matter more (e.g., subtle findings) and how the framework can be extended.

Overall, this is a well-executed validation study with clear practical implications for evaluating reconstruction models in end-to-end clinical pipelines, including fairness considerations. I recommend acceptance.

---

### Decision · Program_Chairs · 2026-02-14

Accept (Poster)